# Treatment Patterns for Gastroesophageal Junction Adenocarcinoma in the United States [note 1]

**DOI:** 10.3390/jcm9113495

**Published:** 2020-10-29

**Authors:** Bradford J. Kim, Yi-Ju Chiang, Prajnan Das, Bruce D. Minsky, Mariela A. Blum, Jaffer A. Ajani, Jeannelyn S. Estrella, Wayne L. Hofstetter, Ching-Wei D. Tzeng, Brian D. Badgwell, Paul F. Mansfield, Naruhiko Ikoma

**Affiliations:** 1Department of Surgical Oncology, The University of Texas MD Anderson Cancer Center, 1400 Pressler Drive, Unit 1484, Houston, TX 77030, USA; bjkim@mdanderson.org (B.J.K.); YChiang1@mdanderson.org (Y.-J.C.); CDTzeng@mdanderson.org (C.-W.D.T.); bbadgwell@mdanderson.org (B.D.B.); pmansfie@mdanderson.org (P.F.M.); 2Department of Radiation Oncology, The University of Texas MD Anderson Cancer Center, 1400 Pressler Drive, Unit 1484, Houston, TX 77030, USA; PrajDas@mdanderson.org (P.D.); BMinsky@mdanderson.org (B.D.M.); 3Department of Gastrointestinal Medical Oncology, The University of Texas MD Anderson Cancer Center, 1400 Pressler Drive, Unit 1484, Houston, TX 77030, USA; mblum@mdanderson.org (M.A.B.); jajani@mdanderson.org (J.A.A.); 4Department of Pathology, The University of Texas MD Anderson Cancer Center, 1400 Pressler Drive, Unit 1484, Houston, TX 77030, USA; JSEstrella@mdanderson.org; 5Department of Thoracic and Cardiovascular Surgery, The University of Texas MD Anderson Cancer Center, 1400 Pressler Drive, Unit 1484, Houston, TX 77030, USA; WHofstetter@mdanderson.org

**Keywords:** chemoradiation, esophagectomy, gastrectomy, GEJ, neoadjuvant therapy

## Abstract

Despite the increasing incidence of gastroesophageal junction adenocarcinoma (GEJA), the optimal treatment strategy for the disease remains unknown. The objective of this study was to describe treatment patterns for GEJA in the United States. The National Cancer Database was searched to identify all patients who underwent resection of the lower esophagus, abdominal esophagus, and/or gastric cardia for GEJA between 2006 and 2016. Patients were grouped by clinical disease stage: early localized (L; T1-2N0), locally advanced (LA; T3-4N0), regional (R; T1-2N+), or regionally advanced (RA; T3-4N+). The search identified 28,852 GEJA patients. The dominant age range was 60–69 years (39%). Most patients were men (85%), and most were white (92%). Most L patients (69%) underwent upfront surgery, whereas most LA, R, and RA patients received neoadjuvant therapy (NAT; 86%, 80%, and 90%, respectively). Among patients who received NAT, 85% received chemoradiotherapy. Adjuvant therapy was relatively uncommon across all groups (15–20%). In the LA, R, and RA groups, overall survival was greater in patients who received NAT compared to upfront surgery (*p* < 0.001). With the exception of patients with early localized node-negative disease, most GEJA patients receive neoadjuvant chemoradiotherapy despite the lack of prospective trials reporting survival benefit over chemotherapy alone.

## 1. Introduction

In the early 1970s, the incidence of gastroesophageal junction adenocarcinoma (GEJA) began rising, increasing by nearly 2.5-fold into the early 1990s, when it stabilized into a plateaued rate [1]. During this time, the advent of the Siewert classification was used to improve treatment plans of adenocarcinomas localized to the gastroesophageal junction (GEJ) [2,3]. Broadly, Siewert defined a GEJ tumor as one with its center within 5 cm proximal and distal to the anatomic gastric cardia [3].

From the past two decades, several large randomized studies have demonstrated significant survival benefit from pre- and/or postoperative therapies in the treatment of gastric [4,5,6,7] and esophageal [8,9] cancer. However, no phase III trial has studied adenocarcinoma localized only to the GEJ independently, and treatment recommendations are based on studies mostly focused on gastric or esophageal cancers [10]. In brief, the phase III Medical Research Council Adjuvant Gastric Infusional Chemotherapy (MAGIC) trial, which included gastric and GEJ cancers, showed the survival benefit of perioperative chemotherapy over surgery alone. The phase III Chemoradiotherapy for Oesophageal Cancer Followed by Surgery Study (CROSS) trial, which included esophageal and GEJ cancers, showed the survival benefit of preoperative chemoradiation therapy over surgery alone. Based on these landmark trials and subsequent prospective studies that reproduced the benefit of such treatment approaches, both preoperative chemotherapy and chemoradiation therapy are recommended in treatment guidelines for GEJA [10]. Currently, there is a paucity of data that specifically investigate the pre- and/or postoperative treatment approaches for GEJA [11,12], and the optimal treatment remains unknown. To address this gap in knowledge, the objective of this study was to describe current treatment patterns for patients who underwent resection for GEJA in this era of increased incidence and use of modern treatment modalities. We therefore examined the current practice pattern in the United States for patients with resectable GEJA by using the largest nationwide cancer database available.

## 2. Methods

### 2.1. Dataset

The National Cancer Database (NCDB) was used for this retrospective cohort analysis. Jointly maintained by the American Cancer Society and American College of Surgeons Commission on Cancer, the NCDB is a prospectively collected hospital-based cancer registry of more than 1500 Commission on Cancer–accredited centers in the United States. This study was approved by the Institutional Review Board of The University of Texas MD Anderson Cancer Center.

#### Patient Selection

The NCDB was searched to identify all patients who underwent resection of the lower esophagus, abdominal esophagus, and/or gastric cardia for GEJA between 2006 and 2016. First, the NCDB was queried for all patients 18 years and older with International Classification of Disease of Oncology, Third Edition (ICD-O-3) topography codes C152–C155 (C152, abdominal esophagus; C153, upper third of the esophagus; C154, middle third of the esophagus; C155, lower third of the esophagus) and C160–C166 (gastric cardia, gastric fundus, gastric body, gastric antrum, gastric pylorus, gastric lesser curvature, and gastric greater curvature). Target histology included: 8140 (adenocarcinoma, not otherwise specified), 8144 (adenocarcinoma, intestinal type), 8145 (adenocarcinoma, diffuse type), and 8490 (signet ring cell adenocarcinoma). All patients who underwent gastrectomy or esophagectomy were identified, and patients without neoadjuvant/adjuvant therapy records were excluded from the analysis. Patients with GEJA were defined as patients who had adenocarcinoma of the abdominal esophagus, lower third of the esophagus, or gastric cardia.

### 2.2. Clinical Variables

Patient demographics including race, gender, age, high school education (% who did not graduate), zip code income, and insurance status were collected. The Charlson Comorbidity Index was included in the analysis. Cancer-related variables included clinical and pathologic TNM (tumor, nodes, and metastases) cancer stage, margin status (R0, R1, R2, or RX), tumor size measured at the time of pathology exam, receipt of lymphadenectomy, adequate lymph node sampling, receipt of neoadjuvant chemotherapy and/or adjuvant chemotherapy, receipt of neoadjuvant and/or adjuvant radiation therapy, mortality, and overall survival. Adequate lymph node sampling was defined as a retrieval of 15 or more lymph nodes (LN ≥ 15) per AJCC guidelines. Neoadjuvant therapy (NAT) was defined as any treatment before surgery, including any combination of chemotherapy and/or radiation therapy. Similarly, adjuvant therapy was defined as any therapy received after surgical resection. Patients with any lymph nodes positive for adenocarcinoma were considered node positive (N+). Patients were grouped by clinical disease stage: early localized (L; T1-2N0), locally advanced (LA; T3-4N0), regional (R; T1-2N+), or regionally advanced (RA; T3-4N+), and practice patterns and treatment outcomes were stratified by these stage groups.

### 2.3. Statistical Analysis

The yearly overall reported number of GEJA cases was calculated as a subset of all yearly adenocarcinoma patients, and the yearly number of resected GEJA was calculated as a subset of all yearly GEJA patients. The Mann–Whitney U-test was used for comparison of continuous data. Categorical data were compared with the Chi-squared test or Fisher’s exact test. Kaplan–Meier estimates and log-rank tests were used to calculate and compare the survival curves, median survival time, and the five-year overall survival for treatment by clinical stage. Multivariable analysis for overall survival in all patients was conducted to investigate factors, including use of NAT, associated with overall survival (OS). A logistic regression model was used to assess the likelihood of NAT vs. surgery first. Cox regression models were used to determine predictors of survival. Interaction terms were evaluated for different survival impacts of NAT by clinical stage to determine appropriate Cox regression models. Covariates with *p* < 0.2 from univariate analysis were retained, then a stepwise selection method was performed to determine the final logistic and Cox regression models with relevant variables. The Hosmer–Lemeshow goodness-of-fit test was applied to test the final model. Odds ratios (OR), hazard ratios (HR), and 95% confidence intervals (95% CI) were estimated for the NAT and survival outcomes. All tests were two-sided. A *p* value < 0.05 was the significance level for all analyses. All data analyses were conducted using SAS Enterprise Guide 7.15 (SAS Inc., Cary, NC, USA).

## 3. Results

Initial analysis of the NCDB identified 239,018 patients from a combined cohort of esophageal and gastric cancer patients. From this group, there were 203,195 patients with squamous cell carcinoma or adenocarcinoma, 72,382 of whom had undergone resection with a gastrectomy or esophagectomy. After excluding patients without a record of neoadjuvant/adjuvant therapy, there remained 71,058 patients who had undergone gastrectomy or esophagectomy for squamous cell carcinoma or adenocarcinoma of any location in these two organs. The distribution for tumor location is shown in Table 1. The majority of patients in this cohort had tumor histology of adenocarcinoma (*n* = 214,281; 83.8%), and the distribution of adenocarcinoma is shown in Table A1.

The total cohort of this study for further analyses included 28,852 GEJA patients, with most patients being male (85.3%) and white (92.1%). This analysis identified 8286 (28.7%) L, 5678 (19.7%) LA, 3327 (11.5%) R, and 11,561 (40.1%) RA GEJA patients. Demographic, clinical, and treatment pattern variables are listed in Table 2. Most patients with clinical stage L GEJA (5710, 68.9%) underwent upfront surgery, whereas most patients with LA (4918, 86.6%), R (2659, 79.9%), and RA (10,512, 90.9%) GEJA received NAT.

The use of NAT constantly increased over the study period in all clinical stages and is represented in Figure 1 (*p* < 0.001). Among patients who received NAT, 85% received chemoradiation, 13% chemotherapy alone, and 2% radiation therapy alone. Neoadjuvant chemoradiation was the most common modality among patients who received NAT across all clinical stages (L: 25.5%, LA: 75.2%, R: 68.1, RA: 78.0%; *p* < 0.001). The use of adjuvant therapy was relatively uncommon (16.4%) across all groups (L: 14.1% LA: 14.8% R: 18.2% RA: 18.3%, *p* < 0.001). Among patients with clinical stage L and LA disease, 1788 (21.6%) and 1717 (30.2%), respectively, were pathologically node positive after surgery. Among patients with clinical N-negative but pathologic N-positive disease, most patients with clinical stage L disease underwent upfront resection (1229, 67.9%) rather than neoadjuvant chemotherapy (18, 1.0%), neoadjuvant radiation (128, 7.1%), and neoadjuvant chemoradiation (435, 24.0%). Most patients with clinical stage LA disease underwent neoadjuvant chemoradiation (1149, 65.4%) before resection. Lastly, among patients with clinical stage LA disease, a greater proportion of surgery-first patients (361, 47.5%) were upstaged to pathological N+ disease compared to those who received NAT (1397, 28.4%; *p* < 0.001). Multivariable logistic regression analyses to identify factors associated with receiving NAT vs. surgery first are represented in Table 3. The use of NAT was associated with esophageal classification (OR 1.57), male gender (OR 1.33), white race (OR 1.27), younger age (OR 1.52; age <50, when compared to age 70–79), and lower Charlson Comorbidity Index (OR 1.43; score 0, when compared to score ≥2), and advanced clinical stage (OR 2.53; T3-T4N+, when compared to T1-2N+).

The median overall survival for GEJA patients undergoing resection was 3.3 years. Median overall survival was significantly greater in patients who received NAT than in patients who underwent upfront surgery in the LA (3.3 years vs. 2.1 years, *p* < 0.001), R (3.4 years vs. 2.4 years, *p* < 0.001), and RA (2.6 years vs. 1.7 years, *p* < 0.001) groups (Figure 2). In the Cox regression model, the interaction terms between the use of NAT and clinical stage were significant (L: HR = 1.53, CI = 1.43–1.65; LA: HR = 0.75, CI = 0.68–0.83; R: HR = 0.86, CI = 0.77–0.96; RA: HR = 0.73, CI = 0.68–0.79, *p* < 0.001), indicating the survival impact of NAT is different by clinical stage. Therefore, Cox regression models were fit for each clinical stage category with only pretreatment factors. The factors associated with worse OS are listed in Table A2 by clinical stage. Use of NAT was associated with improved OS in LA (HR = 0.75, CI = 0.68–0.83, *p <* 0.001), R (HR = 0.87, CI = 0.77–0.98, *p* = 0.017), and RA (HR = 0.70, CI = 0.65–0.75, *p* < 0.001), but not in L (HR = 1.56, CI = 1.45–1.68, *p* < 0.001) patients after adjustments with other factors (Figure 2).

## 4. Discussion

This study represents the largest and most modern analysis of current treatment patterns for resected GEJA in the United States. It identified a large volume of patients undergoing resection for GEJA and showed that the majority of patients with clinical stage LA, R, and RA disease received NAT. Despite the lack of strong evidence showing survival benefit of preoperative chemoradiation therapy over chemotherapy alone for GEJA, most patients who underwent NAT received chemoradiation therapy. A remarkable increase in the use of NAT was observed annually over the decade of this study’s period. As previously reported, nodal upstaging was more frequent in patients with clinical stage L disease, in particular when upfront surgery was performed without NAT.

Based on data from the SEER cancer registry, a remarkable increase in the incidence of esophageal adenocarcinoma at the lower esophageal location was observed from the 1970s to early 2000s, and a substantial proportion of this increase was due to an increase in the incidence of GEJA [13,14]. Although this rise has slowed from a 10% annual increase before 1999 to a 1.6% decline in subsequent years, GEJA continues to be a major public health problem in the United States. The authors of these studies speculated that exposure to the initiating factors (reflux disease or increased body mass index) for esophageal adenocarcinoma have plateaued in exerting maximal potential effect, and that the premalignant transformation of Barrett’s esophagus to cancer could have slowed with screening and surveillance. Our analysis of NCDB showed a persistent high volume of GEJA.

Neoadjuvant therapy for surgically resectable gastric, esophageal, and junctional cancers is well supported by level I evidence [6,7,8,15,16]. As a result, the use of neoadjuvant therapy has rapidly increased for esophageal and gastric cancers over the past two decades [17,18,19]. Our study supplements this information with a more granular analysis specifically for GEJA, in which the treatment strategy is particularly controversial. This study demonstrated that the uptake of NAT has continued to increase in recent years. Despite the lack of high-level evidence supporting the superiority of chemoradiation therapy over chemotherapy alone for GEJA [20,21], a very high proportion of the patients who underwent NAT received chemoradiation therapy in the United States. This could be explained by previous U.S. population studies showing that 82% of gastric cardia cancers are treated by esophagectomy rather than gastrectomy. Therefore, surgeons (likely thoracic surgeons rather than abdominal surgeons) who perform esophagectomy are thought to be more familiar with CROSS trial chemoradiation over MAGIC trial chemotherapy for gastrectomies [22]. The potential benefits of neoadjuvant chemoradiation therapy over chemotherapy alone for GEJA include higher rates of pathological complete response and margin-negative (R0) resection [23], which may result in improved local control [12]. In surgical resection of advanced GEJA, it is often challenging to achieve R0 margins because of the tumor’s narrow and deep anatomical location; therefore, the local effect of chemoradiation therapy may be more beneficial for GEJ tumors than non-GEJ esophageal or gastric cancers. The survival benefit of neoadjuvant chemoradiation therapy vs. chemotherapy alone is being investigated in the international phase III Trial of Preoperative Therapy for Gastric and Esophagogastric Junction Adenocarcinoma (TOPGEAR) trial [24]. The TOPGEAR trial randomly assigned patients with gastric or GEJ adenocarcinomas to receive perioperative chemotherapy (epirubicin/cisplatin/5-fluorouracil) with or without preoperative chemoradiation therapy (45 Gy in 25 fractions with concurrent 5-fluorouracil). Similarly, the Dutch phase II CRITICS-II trial compared preoperative chemotherapy and chemoradiation therapy in gastric and GEJ adenocarcinomas [25]. Although these trials were not specifically designed for GEJA, the results may have implications for the future clinical treatment of GEJA.

There are some limitations of this study. First, the retrospective study design is inherently subject to selection bias. However, this large nationwide cohort provides useful information about the increasing use of neoadjuvant therapy for resected GEJA. To our knowledge, this study is the first nationwide analysis investigating trends in the treatment approach specifically for GEJA. Second, the NCDB lacks granularity, which makes analysis of GEJA challenging, as the tumors are coded in several ways nationally from an esophagus or gastric perspective (lower third of the esophagus vs. abdominal esophagus vs. gastric cardia). We created a dataset that is suggestive of GEJA origin but likely included tumors without involvement of GEJ. And although contributing centers to the NCDB include various types of hospitals that range from community hospitals to academic cancer centers, generalizability of the study results to non-NCDB hospitals may be limited. Next, the survival analyses conducted in this study, particularly in assessment of survival impact of the use of NAT, were severely limited due to selection bias and immortal time bias associated with the use and duration of NAT. Although the models we fit in this study suggested possible survival benefit of NAT in GEJA that support the increased use of NAT as shown in this study, the optimal regimen (particularly comparing chemoradiation therapy vs. chemotherapy only) should be investigated by prospective trials, and results of the above-mentioned trials are awaited. Lastly, this study did not analyze specific methods of resection (esophagectomy or gastrectomy) or surgical approach (open vs. minimally invasive or transthoracic vs. transhiatal), because the purpose of this study was to examine current treatment patterns and sequences of GEJA and not methods of surgical approach. Further, several studies have emphasized the importance of preoperative therapy rather than a surgical approach in oncologic outcomes, specifically overall survival, for patients with this disease [26,27].

## 5. Conclusions

In this study analyzing esophageal and gastric datasets from the NCDB, lower-third esophageal and cardia gastric adenocarcinomas continued to be the main proportion of esophageal and gastric cancers in the United States. The use of NAT for GEJA has steadily increased over the past decade, except for early localized disease, for which upfront surgery is the most common approach. Most patients who underwent NAT received chemoradiation therapy, despite the lack of existing evidence supporting any survival benefit of chemoradiation therapy over chemotherapy alone. Results of ongoing prospective randomized controlled trials may impact future clinical practice for GEJA in the United States.

## Figures and Tables

**Figure 1 jcm-09-03495-f001:**
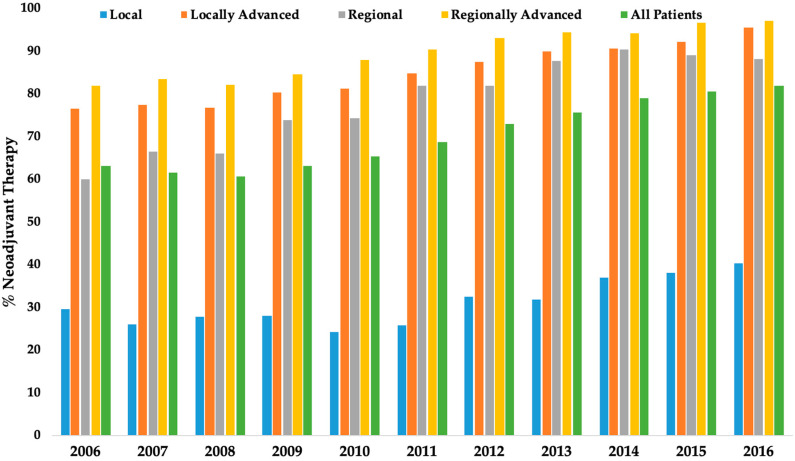
The use of neoadjuvant therapy (chemotherapy, radiation therapy, chemoradiation) by clinical stage from 2006 to 2016 for patients who underwent resection for gastroesophageal junction adenocarcinoma.

**Figure 2 jcm-09-03495-f002:**
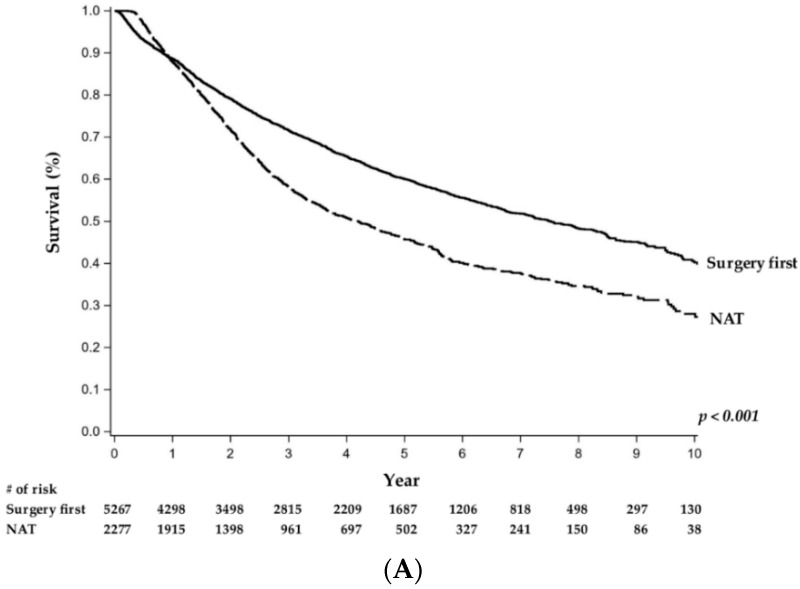
Overall survival analysis comparing patients with gastroesophageal junction adenocarcinoma who received upfront surgery to those who first received neoadjuvant therapy across all clinical stages: (**A**) early localized (T1-2N0), (**B**) locally advanced (T3-4N0), (**C**) regional (T1-2N+), and (**D**) regionally advanced (T3-4N+). Abbreviations: NAT, neoadjuvant therapy. #, number.

**Table 1 jcm-09-03495-t001:** Yearly number of all tumors of the esophagus and stomach and number resected by its anatomic location.

**Esophagus**	**Overall**	**Overall Resected**		**Upper**		**Middle**		**Lower**		**Abdominal**	
	***n***	***n***	**%**	***n***	**%**	***n***	**%**	***n***	**%**	***n***	**%**
2006	9591	2637	27.5	54	2.0	245	9.3	1952	74.0	22	0.8
2007	9817	2650	27.0	46	1.7	229	8.6	2004	75.6	21	0.8
2008	10,061	2654	26.4	40	1.5	215	8.1	2004	75.5	22	0.8
2009	10,308	2615	25.4	37	1.4	206	7.9	1987	76.0	11	0.4
2010	10,033	2455	24.5	37	1.5	226	9.2	1822	74.2	14	0.6
2011	10,205	2531	24.8	35	1.4	255	10.1	1902	75.1	14	0.6
2012	10,578	2500	23.6	38	1.5	238	9.5	1903	76.1	9	0.4
2013	10,999	2722	24.7	42	1.5	262	9.6	2088	76.7	5	0.2
2014	11,095	2648	23.9	33	1.2	251	9.5	2039	77.0	10	0.4
2015	11,524	2651	23.0	40	1.5	204	7.7	2101	79.3	7	0.3
2016	11,471	2627	22.9	33	1.3	235	8.9	2061	78.5	7	0.3
**Gastric**	**Overall**	**Overall Resected**		**Cardia**		**Proximal**		**Distal**		**Lesser/Greater Curvature**	
	***n***	***n***	**%**	***n***	**%**	***n***	**%**	***n***	**%**	***n***	**%**
2006	10,955	5101	46.6	1610	31.6	549	10.8	1229	24.1	649	12.7
2007	11,332	5146	45.4	1513	29.4	553	10.7	1291	25.1	711	13.8
2008	11,485	5090	44.3	1473	28.9	579	11.4	1294	25.4	690	13.6
2009	12,110	5225	43.1	1610	30.8	583	11.2	1354	25.9	655	12.5
2010	12,532	5230	41.7	1758	33.6	561	10.7	1281	24.5	708	13.5
2011	13,047	5410	41.5	1887	34.9	562	10.4	1337	24.7	732	13.5
2012	13,317	5391	40.5	1814	33.6	520	9.6	1371	25.4	696	12.9
2013	13,845	5460	39.4	1927	35.3	653	12.0	1401	25.7	636	11.6
2014	13,964	5409	38.7	1924	35.6	697	12.9	1315	24.3	616	11.4
2015	13,911	5226	37.6	1951	37.3	643	12.3	1232	23.6	604	11.6
2016	13,622	4976	36.5	1817	36.5	615	12.4	1191	23.9	584	11.7

Abbreviations: *n*, number. Overall resected %: proportion of patients who underwent resection among overall esophageal or gastric cancer patients for each year. Esophageal anatomical locations (upper, middle, lower, abdominal) and gastric anatomical locations (cardia, proximal, distal, lesser/greater curvature) percentages are the proportion of resected cases at each specified location among overall resected cases for each year.

**Table 2 jcm-09-03495-t002:** Demographic, clinical, and treatment variables for patients who underwent resection of gastroesophageal junction adenocarcinoma tumors by clinical stage.

	Total		T1/T2 N0		T3/T4 N0		T1/T2 N+		T3/T4 N+		*p*
	*n* = 28,852	%	*n* = 8286	%	*n* = 5678	%	*n* = 3327	%	*n* = 11,561	%	
Gender											<0.001
Male	24,611	85.3	6900	83.3	4818	84.9	2878	86.5	10,015	86.6	
Age											<0.001
<50	2622	9.1	581	7.0	508	9.0	303	9.1	1230	10.6	
50–59	7128	24.7	1809	21.8	1308	23.0	826	24.8	3185	27.6	
60–69	11,144	38.6	3102	37.4	2219	39.1	1344	40.4	4479	38.7	
70–79	6862	23.8	2324	28.1	1414	24.9	746	22.4	2378	20.6	
≥80	1096	3.8	470	5.7	229	4.0	108	3.3	289	2.5	
Race											0.758
White	26,572	92.1	7607	91.8	5251	92.5	3057	91.9	10,657	92.2	
Black	690	2.4	211	2.6	126	2.2	79	2.4	274	2.4	
Hispanic	836	2.9	232	2.8	156	2.8	105	3.2	343	3.0	
Asian/Pacific Islander	372	1.3	118	1.4	73	1.3	34	1.0	147	1.3	
Other	161	0.6	50	0.6	30	0.5	25	0.8	56	0.5	
Unknown	221	0.8	68	0.8	42	0.7	27	0.8	84	0.7	
Anatomical Location											<0.001
Abdominal esophagus	80	0.3	27	0.3	9	0.2	10	0.3	34	0.3	
Lower esophagus	15,380	53.3	4239	51.2	3006	52.9	1833	55.1	6302	54.5	
Gastric cardia	13,392	46.4	4020	48.5	2663	46.9	1484	44.6	5225	45.2	
Organ											<0.001
Esophagus	15,460	53.6	4266	51.5	3015	53.1	1843	55.4	6336	54.8	
Stomach	13,392	46.4	4020	48.5	2663	46.9	1484	44.6	5225	45.2	
Facility Type											<0.001
Community Program	9619	33.3	2663	32.1	2120	37.3	1089	32.7	3747	32.4	
Academic Program	19,233	66.7	5623	67.9	3558	62.7	2238	67.3	7814	67.6	
Comorbidity Score											
0	20,113	69.7	5431	65.5	3999	70.4	2355	70.8	8328	72.0	
1	6566	22.8	2084	25.2	1274	22.4	741	22.3	2467	21.3	
≥2	2173	7.5	771	9.3	405	7.1	231	6.9	766	6.6	
Education											0.017
≥29%	3202	11.1	909	11.0	694	12.2	378	11.4	1221	10.6	
<29%	24,855	86.2	7152	86.3	4810	84.7	2852	85.7	10,041	86.9	
Unknown	795	2.8	225	2.7	174	3.1	97	2.9	299	2.6	
Zip Code Income											0.184
≥$46,000	12,180	42.2	3461	41.8	2341	41.2	1400	42.1	4975	43.0	
<$46,000	15,879	55.0	4598	55.5	3163	55.7	1830	55.0	6288	54.4	
Unknown	793	2.8	224	2.7	174	3.1	97	2.9	298	2.6	
Insurance											<0.001
Private	13,387	46.4	3378	40.8	2486	43.8	1645	49.4	5878	50.8	
Not Insured	504	1.8	133	1.6	98	1.7	47	1.4	226	2.0	
Government’s Plan	14,520	50.3	4616	55.7	3006	52.9	1603	48.2	5295	45.8	
Unknown	441	1.5	159	1.9	88	1.6	32	1.0	162	1.4	
Margin											<0.001
R0	26,114	90.5	7702	93.0	5079	89.5	3035	91.2	10,298	89.1	
R1	1101	3.8	207	2.5	253	4.5	111	3.3	530	4.6	
R2	59	0.2	11	0.1	12	0.2	6	0.2	30	0.3	
RX	1578	5.5	366	4.4	334	5.9	175	5.3	703	6.1	
Pathologic T											<0.001
T0	3533	12.3	546	6.6	710	12.5	535	16.1	1742	15.1	
T1	6536	22.7	3961	47.8	644	11.3	822	24.7	1109	9.6	
T2	4217	14.6	1249	15.1	776	13.7	642	19.3	1550	13.4	
T3	8577	29.7	1034	12.5	2344	41.3	579	17.4	4620	40.0	
T4	462	1.6	53	0.6	148	2.6	32	1.0	229	2.0	
TX	5527	19.2	1443	17.4	1056	18.6	717	21.6	2311	20.0	
Tumor Size											<0.001
≤5 cm	15,334	53.2	5586	67.4	2813	49.5	1796	54.0	5139	44.5	
>5–10 cm	4182	14.5	572	6.9	839	14.8	433	13.0	2338	20.2	
>10–15 cm	300	1.0	34	0.4	54	1.0	26	0.8	186	1.6	
>15 cm	222	0.8	91	1.1	42	0.7	25	0.8	64	0.6	
Unknown	8814	30.6	2003	24.2	1930	34.0	1047	31.5	3834	33.2	
Lymphadenectomy (LN)											<0.001
<15 nodes	12,918	44.8	3910	47.2	2644	46.6	1479	44.5	4885	42.3	
≥15 nodes	14,086	48.8	3855	46.5	2643	46.6	1652	49.7	5936	51.4	
Unknown	171	0.6	24	0.3	45	0.8	16	0.5	86	0.7	
LN Positive											<0.001
Negative	15,727	54.5	5971	72.1	3566	62.8	1457	43.8	4733	40.9	
Positive	11,224	38.9	1788	21.6	1717	30.2	1663	50.0	6056	52.4	
No LN examined	1677	5.8	497	6.0	346	6.1	180	5.4	654	5.7	
Unknown	224	0.8	30	0.4	49	0.9	27	0.8	118	1.0	
Pathological N stage											<0.001
N0	17,246	59.8	6461	78.0	3883	68.4	1608	48.3	5294	45.8	
N+	11,462	39.7	1810	21.8	1758	31.0	1703	51.2	6191	53.6	
NX	144	0.5	15	0.2	37	0.7	16	0.5	76	0.7	
Neoadjuvant Treatment											<0.001
Surgery first	8187	28.4	5710	68.9	760	13.4	668	20.1	1049	9.1	
Chemotherapy alone	2659	9.2	412	5.0	575	10.1	340	10.2	1332	11.5	
XRT alone	340	1.2	51	0.6	72	1.3	53	1.6	164	1.4	
Chemoradiation	17,666	61.2	2113	25.5	4271	75.2	2266	68.1	9016	78.0	
Adjuvant Treatment											<0.001
None	24,127	83.6	7120	85.9	4840	85.2	2721	81.8	9446	81.7	
Chemotherapy alone	2644	9.2	515	6.2	474	8.4	329	9.9	1326	11.5	
XRT alone	443	1.5	56	0.7	108	1.9	48	1.4	231	2.0	
Chemoradiation	1638	5.7	595	7.2	256	4.5	229	6.9	558	4.8	
Treatment Sequence											<0.001
Surgery only	6079	21.1	4812	58.1	450	7.9	381	11.5	436	3.8	
Neoadjuvant	18,048	62.6	2308	27.9	4390	77.3	2340	70.3	9010	77.9	
Adjuvant	2108	7.3	898	10.8	310	5.5	287	8.6	613	5.3	
Neoadjuvant + adjuvant	2617	9.1	268	3.2	528	9.3	319	9.6	1502	13.0	

Abbreviations: LN, lymph node; T, tumor; TX, tumor not assessed; N, nodes; N+, node positive; NX, nodes not assessed; XRT, radiation therapy.

**Table 3 jcm-09-03495-t003:** Logistic regression analysis for factors associated with receiving NAT vs. surgery first.

*n* = 28,852	OR	95% CI		*p*
Organ				<0.001
Esophagus vs. Stomach	1.57	1.47	1.68	
Gender				<0.001
Male vs. Female	1.33	1.22	1.46	
Age (ref. <50)				<0.001
50–59	1.02	0.90	1.17	
60–69	0.95	0.84	1.08	
70–79	0.66	0.57	0.76	
≥80	0.15	0.12	0.18	
Race (ref. Non White)				<0.001
White	1.27	1.13	1.44	
Unknown	0.76	0.53	1.09	
Insurance (ref. Private)				0.014
Not Insured	1.09	0.84	1.42	
Government plan	0.92	0.85	0.99	
Unknown	0.72	0.56	0.93	
Facility Type				0.004
Academic vs. Community	0.90	0.84	0.97	
Comorbidity score (ref. ≥2)				<0.001
0	1.43	1.27	1.61	
1	1.18	1.04	1.34	
Clinical stage (ref. T3-4N+)				<0.001
T1-2N0	0.04	0.04	0.05	
T3-4N0	0.69	0.63	0.77	
T1-2N+	0.40	0.35	0.44	

Abbreviations: NAT, neoadjuvant therapy; OR, odds ratio; ref., reference; CI, confidence Interval.

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
