# Peer review of "Treatment Patterns for Gastroesophageal Junction Adenocarcinoma in the United States†"

_jcm, 2020, doi:10.3390/jcm9113495_

Round 1

Reviewer 1 Report

I am now satisfied with the revisions to the manuscript. Congratulations to the authors on their work. 

Reviewer 2 Report

Authors addressed all points I raised. I have no further concerns.

This manuscript is a resubmission of an earlier submission. The following is a list of the peer review reports and author responses from that submission.

Round 1

Reviewer 1 Report

Review: Treatment patterns for gastroesophageal junction adenocarcinoma in the United States

Summary

This study was done to describe treatment patterns for GEJA in the United States. The National Cancer Database was searched to identify all patients who underwent resection of the lower oesophagus, abdominal oesophagus, and/or gastric cardia for GEJA between 2006 and 2016. Patients were grouped by clinical disease stage: early localized (L; T1-2N0), locally advanced (LA; T3-4N0), regional (R; T1-2N+), or regionally advanced (RA; T3-4N+). The search identified 28,852 GEJA patients.

Among patients who received NAT, 85% received chemoradiotherapy. Adjuvant therapy was relatively uncommon across all groups. In the LA, R, and RA groups, overall survival was greater in patients who received NAT compared with upfront surgery.

The paper is clear and well written. I have a number of comments which are listed below -

Major Comments

  1. Does the study truly address the knowledge gap that it intends to? (page 2 line 54) The study could provide more information to address the role of NAT among patients with GEJA. For example, the authors do not provide any analysis of CRT vs chemotherapy only. Furthermore, the authors do not provide any subgroup analysis of patient factors associated with survival benefit following NAT, e.g. patient age, patient comorbidity etc. It is completely unsurprising that the use of NAT has increased over time as the evidence base supporting its use has increased. The data provided in this study could be used to provide a useful analysis of factors associated with survival benefit following NAT, and CRT vs chemo alone, but the authors have not provided this. Overall my main criticism is that the analysis is too basic and could provide much more detail to inform clinical decision making.
  2. The authors conclude that most patients with GEJA receive neoadjuvant CRT rather than chemotherapy alone despite a lack of evidence for a benefit of added radiation. However trials to date, with the exception of FLOT, demonstrate superior histopathologic response following neoadjuvant chemoradiation as compared with chemotherapy alone. Ongoing trials such as NeoAEGIS and ESOPEC, as well as TOPGEAR, aim to provide a head to head comparison regarding this question. The authors display a bias towards chemotherapy alone which ought to be justified within the paper if presented in this manner.
  3. Table 1 and 2 should be moved to supplemental material or combined into a single table detailing only the annual breakdown of treatment modalities for junctional adenocarcinoma- data for SCC and gastric cancer should be removed for clarity.
  4. The study does not include a multivariable survival analysis to account for the effect of confounders with respect to the survival impact of neoadjuvant therapy. This is a major weakness of the current paper and should be addressed by the authors in response to the review.
  5. The risk is that the dataset has failed to adjust for other unmeasured confounders. For example, open vs MIE, transthoracic vs transhiatal. This should be mentioned in the limitations section.
  6. European data often show earlier stage cancers for AEG1 vs AEG2 and AEG3 – this is likely due to increased early dysphagia with esophageal tumours and also the widespread adoption of Barrett’s surveillance programmes in Europe. The present USA data do not demonstrate this trend. The authors may comment on this? What is the policy regarding Barrett’s surveillance in the registered cancer centres?
  7. What is the indication for upfront surgery in locally advanced disease? A significant proportion of patients in the present cohort were treated with upfront surgery in the context of >cT2 Nany disease – is this in keeping with NCI recommendations?
  8. Figure 2A – in the text, the authors indicate that the benefit of NAT was only seen in LA, R and RA groups, however the Figure indicates that patients with L (T1-2N0) disease also benefit from NAT (P<0.0001). Please clarify??

Minor Comments

  1. Page 2 line 45 – the authors suggest that treatment recommendations are based on studies of gastric or esophageal cancers, however the CROSS trial included only esophageal cancers – please revise. In addition the CROSS trial did provide details of outcomes for patients with adenocarcinoma distinct from the entire cohort and can therefore be used to inform management of this group.
  2. Page 2 line 52 - “Currently, there is a paucity of data that specifically investigate the pre- and/or postoperative treatment approaches for GEJA” – suggest change to “there are a paucity of data to…”
  3. The study is limited by the use of national registry data, which despite providing a large sample size lack granularity – there may be factors which influence outcome which cannot be accounted for in the analysis. Furthermore, these data only include outcomes from accredited centres – it is likely that a significant proportion of patients undergo treatment outside of these centres, and outcomes in that cohort are not demonstrated here. While this may provide data from high level academic institutions, the generalisability of these data in a broader context is uncertain. This should be further acknowledged in the limitations section.
  4. Page 2, line 82, “receipt of lymph node examination” – please clarify – do you mean pathologic examination of nodes? Do you mean lymphadenectomy?
  5. Page 3 line 93 – “incidence” – the authors describe incidence figures, however the database used only includes registered cancer centres and is not a population based database. How do the authors know that the figures included reflect changes in incidence and not changes in referral patterns?
  6. Line 179 reference?
  7. “As previously reported, nodal upstaging was more frequent in patients with clinical stage L disease in particular when upfront surgery was performed without NAT.” – please provide more detail regarding this statement in the results – this is an important finding of the study

Reviewer 2 Report

The authors elucidated the treatment trends of gastroesophagal junction adenocarcinoma (GEJA) in the US using the National Cancer Database (NCDB). For localized diseases upfront surgery was dominantly employed, but for more advanced cancer neoadjuvant (NAT) chemotherapy or chemoradiation (85%) was chosen. In patients with advanced stage, overall survival was better in NAT than in upfront surgery. I would like to congratulate the authors on such a clear real data provision. It would be really informative and interesting for global readers. There are a few minor questions to the authors.

  1. Why were the patients without adjuvant/neoadjuvant chemotherapy excluded from the analysis? (in the “Methods” section)
  2. Is adequate lymphadenectomy defined as “15 or more” or “16 or more”? (in the “Table 3”)